# Bioactive Polyphenols Modulate Enzymes Involved in Grapevine Pathogenesis and Chitinase Activity at Increasing Complexity Levels

**DOI:** 10.3390/ijms20246357

**Published:** 2019-12-17

**Authors:** Antonio Filippi, Elisa Petrussa, Francesco Boscutti, Marco Vuerich, Urska Vrhovsek, Zohreh Rabiei, Enrico Braidot

**Affiliations:** 1Department of Agriculture, Food, Environmental and Animal Sciences, Plant Biology Unit, University of Udine, via delle Scienze 91, 33100 Udine, Italy; antoniofilippi@yahoo.it (A.F.); elisa.petrussa@uniud.it (E.P.); francesco.boscutti@uniud.it (F.B.); marco.vuerich@gmail.com (M.V.); rabiei.zohreh@uniud.it (Z.R.); 2Department of Food Quality and Nutrition, Research and Innovation Centre, Fondazione Edmund Mach (FEM), Via Mach 1, 38010 San Michele all’Adige (TN), Italy; urska.vrhovsek@fmach.it; 3Department of Animal Biotechnology, Faculty of Agricultural Biotechnology, National Institute of Genetic Engineering and Biotechnology, P.O. Box 14965-161 Tehran, Iran

**Keywords:** *Ailanthus altissima*, chitinase, plant extracts, *Olea europaea*, pathogenesis-related protein, polyphenols, TOPSIS, *Vitis vinifera*

## Abstract

The reduction of synthetic chemistry use in modern viticulture relies on either the biological control of microorganisms or the induction of pathogenesis-related proteins. In the present study, the effects of hydro-alcoholic plant extracts (PEs) (i.e., by-products of *Vitis vinifera* L., leaves of *Olea europaea* L. and *Ailanthus altissima* (Mill.) Swingle) were tested on purified enzymes activity involved in plant-pathogen interactions. The polyphenolic composition was assayed and analyzed to characterize the extract profiles. In addition, suspension cell cultures of grapevine were treated with PEs to study their modulation of chitinase activity. Application of grape marc’s PE enhanced chitinase activity at 4 g L^−1^. Additionally, foliar treatment of grape marc’s PE at two doses (4 g L^−1^ and 800 g L^−1^) on grapevine cuttings induced a concentration-dependent stimulation of chitinase activity. The obtained results showed that the application of bioactive compounds based on PEs, rich in phenolic compounds, was effective both at in vitro and ex/in vivo level. The overall effects of PEs on plant-pathogen interaction were further discussed by applying a multi-criteria decision analysis, showing that grape marc was the most effective extract.

## 1. Introduction

The prevention of pathogen infection in vineyards is a big challenge since the grapevine is frequently exposed to several diseases [1].

During the last half-century, synthetic chemicals have been largely utilized in viticulture, thanks to their high efficacy, easy application, and low cost [2]. Their specific design, addressed to fight harmful plant pathogens, presents hazards for the environment due to their potential effects upon non-target organisms, including humans. Chemical residues are liable to remain on the plant or within its tissues for the following treatments [2].

In order to reduce the environmental impact of traditional agricultural pest management, alternative use of bioagents, obtained from microorganisms or agricultural/plant wastes, is a new nature- and human-friendly strategy representing a promising sustainable pathogen-controlling tool. Such treatments can trigger grapevine plants to secrete specific secondary metabolites or to enhance the activity of enzymes able to counteract the pathogen attack, improving the plant immunity [3,4].

A unique definition of bioagents is not universally accepted. In fact, these compounds, since they are obtained from the different starting material, exhibit extremely different chemical composition and modulate the most diverse physiological functions [5]. The commonly used definition regards the nutraceutical properties of bioagents based on plant extracts (PEs), rich in secondary metabolites, and attains specifically the effect of such compounds on animal metabolism and health [6]. In the food sciences, bioactive compounds are considered as «phytochemicals […] that are capable of modulating metabolic processes and resulting in the promotion of better health [...] they exhibit beneficial effects, such as antioxidant activity, inhibition or induction of enzymes, inhibition of receptor activities, and induction and inhibition of gene expression [6]. Nevertheless, it is reasonable to extend this definition not only to animals as target organisms but also to living organisms in general, including plants themselves [7].

Therefore, remarkable attention has been recently paid on plant extracts (PEs) properties and how plant perceives them as elicitors of defense against pathogens [4]. Particularly, the search for PEs, exerting either direct antifungal activity or stimulation of plant immune system, has recently attracted the interest of the scientific community [8], and many studies have investigated their molecular mechanism of induction [8,9,10,11].

When the plant perceives a molecule as an elicitor, a network of different signal cascades activates the defense/immunity mechanism response. An operative immune system involves a wide array of gene expression modifications [12], mostly related to the induced resistance in the infected cells at two levels, localized acquired resistance and whole plant systemic acquired resistance [9]. The plant defense mechanisms work effectively synthesizing: (i) pathogenesis-related (PR) proteins (like chitinases), able to directly damage pests; (ii) antimicrobial compounds, such as phytoalexins; and (iii) enzymes (like peroxidases) involved in the cell wall reinforcement and reactive oxygen species formation [9]. Furthermore, the bioactive compounds present in PEs possess an intriguing role in direct modulation of defensive enzymes, as recently claimed by a study on a grapevine class IV chitinase, demonstrating its modulation by two common flavonoids, quercetin and catechin [13].

Chitinases are chitin-hydrolyzing enzymes with low molecular weight (ranging from 25 to 40 kDa) [14], playing a role in the activation of host defense systems. They are classified into two protein families, PR-3 and PR-4, including seven classes, operating on pathogen cell wall or phytophagous exoskeleton as their specific target site [3,4]. In plants, PR chitinase synthesis is induced upon the attack of phytopathogens and confers to the plant self-defense ability against such pathogens [15].

Some studies have reported the over-expression of chitinase in different transgenic plants (strawberries, *Phaseolus vulgaris*, *Vitis vinifera* cultivars) [3], which conferred, in some cases, resistance to major diseases [16,17]. In particular, grapevine resistant cultivars exhibit higher PR expression or activation of the biosynthetic pathway of defense metabolites if compared to susceptible clones [18,19,20]. However, since transgenic grapevines are forbidden in several countries, Aziz and coworkers [21] proposed an alternative strategy for controlling pathogens, relying on the application of elicitors for the activation of plant defense response. Consistently, it is widely accepted that the effects on grape quality exerted by treatments with elicitors or bio-stimulants depend essentially on the number of applications, season, and variety [22]. In the narrowest sense, the direct grapevine plant exposure to PEs leads to the defense mechanism enhancement via stimulation of PR-3 and -4 at transcriptomic/translational levels [3], despite there is evidence that their effects would be weaker when compared to conventional fungicides.

At the same time, any strategy at the plant/pathogen interaction level that is capable to stimulate or induce the expression of PR proteins, as well as to inhibit enzymes involved in the pathogenic process (such as cellulase, pectinase, laccase, or amylase) [8], could be of large interest in agricultural application.

This study aimed to evaluate the efficacy of PEs from different plant species/agro-industrial wastes on the modulation of chitinase enzymes at three different levels of analysis: in the purified enzyme (in vitro), in grapevine cell cultures (ex vivo), and in plants (in vivo). Moreover, the modulation of PEs on pathogen enzymes generally involved in plant-pathogen interaction was also investigated.

## 2. Results

### 2.1. HPLC-MS Quantitative Analysis of the Assayed PEs

Polyphenols play a crucial role in plant response to several biotic and abiotic stresses. For this reason, we measured by HPLC-MS the polyphenolic composition of four PEs obtained from *Ailanthus* leaves, grape marc, grape stalks, and olive leaves (Table 1). The analytical characterization evidenced that phenolic acids and the derived esters were mainly present in both grape PEs and *Ailanthus* leaves and mostly represented by gallic and ellagic acids, besides to the high concentration of caftaric acid found only in stalk PE. On the contrary, olive leaves PE exhibited a scarce amount of phenolic acids, except for protocatechuic acid.

Regarding stilbenoids, they were scarcely represented in all tested PEs, being grape stalk the only PE showing a high content of *t*-resveratrol. HPLC-MS analysis discriminated six different hydroxystilbenes, such as resveratrol and its derivatives piceid, piceatannol, and isorhapontin. These compounds were enriched only in PEs derived from grape stalks and marc, apart from piceid, identified also in *Ailanthus* PE. The order of magnitude of concentration for the latter compounds was quite low if compared to those observed in other phenolic classes (in most cases <1 g L^−1^), except for *t*-resveratrol, which reached the content of 2.36 g L^−1^ in stalk PE. In fact, the latter extract was derived from the lignified stalk of the grape cluster and was, therefore, characterized by a high portion of lignin and tannins on its biomass [23], when compared to the other PEs.

The most represented polyphenols were flavonols, present at high concentrations in all the analyzed PEs, although the different extracts showed high specificity in their composition. In particular, *Ailanthus* leaves had a very high concentration of quercetin-3-glycosides (both as glucose, galactose-, and rhamnose-substituted) and rutin, sharing a good similarity with olive leaves and grape stalk PE profiles. Additionally, they contained also kaempferol-3-rutinoside. On the contrary, unsubstituted quercetin was not detectable in *Ailanthus* PE, while an appreciable amount was measurable in the other PEs. In addition, quercetin-glucuronide derivatives and syringetin-glycosides were also found only in the marc and stalk PEs.

Concerning all other flavonoid compounds, the analysis revealed that both grape marc and stalks exhibited a very high concentration of flavanols, such as catechin, epicatechin, procyanidin B2 + B4, procyanidin B1, and procyanidin B3, the former being present also in *Ailanthus*. Indeed, grape stalk PE was highly enriched in flavonols, associated with a high content in gallocatechin, epigallocatechin, and in the flavanonol taxifolin. Moreover, leaf extracts from olive and *Ailanthus* were characterized by flavones, such as luteolin and its glucose-substituted derivative.

### 2.2. In Vitro Modulation of Plant Defense-Related Enzyme Activities: Chitinase and Peroxidase

Figure 1A shows that chitinase hydrolytic activity was significantly affected by PEs, depending on extract type and concentration. In particular, the lowest concentrations (0.4 g L^−1^) of all PEs, apart from *Olea*, caused a significant stimulation of the chitinase activity compared to the control. However, grape marc and stalk PEs induced a stimulation even at 4 g L^−1^ concentration. Among the studied PEs, grape marc was the most effective in stimulation of chitinase activity, causing almost a 1.5-fold increase (approx. 1800 RFU µg prot^−1^ min^−1^) at 4 g L^−1^ and 1.2-fold at 0.4 g L^−1^, respectively. Conversely, either a partial or total enzyme inhibition was observed at the highest concentration of 40 g L^−1^ for all PEs. In particular, a 90% reduction of basal chitinase activity was measured in the case of treatments with *Ailanthus* and olive leaves PEs.

Figure 1B shows the effect of PEs on horseradish peroxidase activity, a well-recognized PR protein. An effective stimulation on peroxidase activity was observed at 0.4 g L^−1^ in the case of *Ailanthus* PE (2-fold increase when compared to the control). The extracts of *Ailanthus* leaves, grape marc, and grape stalks strongly suppressed peroxidase activity at higher concentrations (4 and 40 g L^−1^), while olive leaves, at any applied concentrations, did not affect peroxidase activity.

Considering all data, it is worth noting that only *Ailanthus* PE was able to stimulate both chitinase and peroxidase to some extent.

### 2.3. In Vitro Modulation of Purified Pathogen-Related Enzymes Activities

The in vitro stimulation/inhibition of purified fungal-related enzymes’ activities (i.e., amylase, cellulase, laccase, and pectinase) was investigated upon incubation with the aforementioned hydro-alcoholic PEs. It is well worth noting that the more pathogen-related enzyme was suppressed, the more PEs were successful in controlling the pathogen attack. As it is shown in Figure 2, most of the PEs at different concentrations affected amylase activity, depending on the extract and concentration. Although grape stalks and olive leaves extracts significantly reduced the amylolytic activity at each applied concentration (Figure 2A), *Ailanthus* leaf extract was effective only at 40 g L^−1^. Instead, grape marc extracts stimulated amylase activity at 0.4 and 4 g L^−1^, albeit at 40 g L^−1^ amylase activity was halved compared to the control, providing an effective inhibition.

In the case of cellulase (Figure 2B), its activity was remarkably decreased in the case of 40 g L^−1^ of grape marc, achieving almost 70% of inhibition when compared to the control. The same effect was observed on cellulase activity with the application of grape stalks extract. On the contrary, no inhibition was detected when *Ailanthus* leaves and olive leaves extracts were applied, but rather a stimulatory effect was measured upon the application of the latter at the highest concentration.

The extracts of grape marc and grape stalks drastically inhibited laccase activity over 4 and 40 g L^−1^ concentration (Figure 2C), being grape marc the most effective treatment, since 0.4 g L^−1^ strongly reduced the activity by more than 50% with respect to the control. Other PEs, such as those obtained from *Ailanthus* and olive leaves, caused such inhibitory effect at 4 g L^−1^ concentration.

Pectinase activity (Figure 2D) was also modulated by all PEs in a concentration-dependent mode. The lowest concentrations effectively reduced the activity compared to the control. Moreover, at 40 g L^−1^ PE, the activity was significantly enhanced by all PEs, apart from the grape marc.

### 2.4. Ex Vivo Modulation of Chitinase Activity in V. vinifera cv. Cabernet Sauvignon Cell Suspension Cultures

The potential of PEs in triggering grapevine chitinase was determined by the pre-incubation of PEs with *V. vinifera* cv. Cabernet Sauvignon cell suspension cultures for 48 h, at a concentration considered moderate (4 g L^−1^), based on the previous results obtained with purified enzymes (see Figure 1). The chitinase activity of cell supernatant was found to be significantly affected by PEs of both grape marc and olive leaves (Figure 3A). In particular, the chitinase stimulation induced by marc extract reached a 1.5-fold increase if compared to the control. Conversely, the treatment with the extract of olive leaves significantly inhibited chitinase activity. Moreover, the extracts of *Ailanthus* leaves and grape stalks did not provide any significant change in the enzyme activity in comparison with the control.

### 2.5. Cell Death of V. vinifera cv. Cabernet Sauvignon Cell Suspension Cultures

Cell death measurement of PEs-treated grapevine cells was carried out to verify the phytotoxicity of PEs (Figure 3B). In all PE-treatments, an increase in cell death was detected, ranging from 21 ± 2% (grape stalks) to 28 ± 19% (*Ailanthus* leaves), if compared to the physiological level (13 ± 3%) measured in non-treated cells. Nevertheless, only the olive leaves’ PE treatment caused a significant cytotoxic effect, which was significantly different from those exerted by the other treatments.

### 2.6. In Vivo Time-Course Modulation of Chitinase Activity Stimulation

Due to the in vitro and ex vivo observations obtained in the previous experiments, a foliar application of grape marc extract on grapevine cuttings was performed in controlled greenhouse conditions at two distinct concentrations of 4 and 800 g L^−1^, respectively (Figure 4).

None of the described treatments caused cytotoxic symptoms at a visual inspection of the leaf (data not shown). Time-course evaluation of leaf chitinase activity showed that both concentrations significantly affected grapevine enzymatic activity. The lower concentration increased chitinase activity over time, starting from 24 and 48 h after treatment (Figure 4A). At the highest activation, the enzyme catalysis was increased up to a two-fold stimulation compared to the control. Differently, the 800 g L^−1^ treatment caused an earlier and stronger stimulation of chitinase, in comparison to the lower dose, causing a seven-fold stimulation of chitinase even at 8 h after PE application (Figure 4B). A further enhancement of the enzyme activity was also detected after 48 h of incubation (nine-fold compared to the control).

### 2.7. A Synthetic Assessment of the In Vitro Effectiveness of the Considered Plant Extracts

Among the applied PEs, grape marc resulted in the best solution, maximizing the overall response of plant- and pathogen-related enzymes, as it was the most effective in stimulating plant defensive enzymes and inhibiting pathogen catalytic activities (Table 2; highest mean value in the right column). In particular, the 40 g L^−1^ fresh weight (FW) of grape marc concentration showed the highest absolute value (0.64). Considering the PE concentration, 4 g L^−1^ FW resulted in the most suitable concentration on the average. In the case of the latter concentration, the stalk PE was found to exhibit the highest score. These effects could be tentatively imputable to their different polyphenolic composition and consequently to a different best concentration where treatments exerted their desired effect. On the contrary, the weakest effective PE appeared to be *Ailanthus* leaves since it reached a more negative score, both as a mean or absolute value.

### 2.8. Principal Component Analysis (PCA) Applied on Polyphenolic Profile of the Tested PEs and Assayed Enzymatic Activities

The results of the PCA applied to the polyphenolic profile of the PEs suggested a clear separation between the considered assays (Figure 5). The first two axes explained 83% of the total variance. The first principal component axis (PC1) manly discriminated between grape stalks and olive and *Ailanthus* leaves, whereas the second (PC2) clearly distinguished the composition of grape marc from all other PEs. Along the PC1, the majority of stilbenoids decreased (e.g., *t*-resveratrol, piceatannol, *t*-piceid, *cis*-piceid, isorhapontin, mainly associated with grape stalk PE). The only stilbene that differed along the PC2 component was instead *cis*-resveratrol, which was uniquely associated with grapevine marc. On the other hand, protocatechuic acid and luteolin (retrievable mainly in olive PE) increased along PC1. The PC2 was mainly explained by the increase of benzoates and cinnamates (e.g., vanillic acid, gallic acid, 3,5-dihydroxy-benzoic acid, protocatechuic acid, methyl gallate, *p*-coumaric acid, caffeic acid, ferulic acid) and flavonols (kaempferol, quercetin, taxifolin, quercetin-3-Rha, myricitrin, quercetin-3-Glc + quercetin-3-Gal, isorhamnetin-3-Glc, syringetin-3-Glc+syringetin-3-Gal) compounds, and some flavan-3-ols, such as epicatechin, catechin gallate, and procyanidin B2 + B4, were strongly associated to grape marc’s PE.

## 3. Discussion

Thanks to their sessile nature, plants have demonstrated to be extremely rich sources of chemical compounds with numberless biological properties [26]. All the estimated 3 million different plant species on earth possess more than 100,000 different compounds, mainly part of a chemically heterogeneous group called secondary metabolites [27], used to face different biotic [28] and abiotic stress [29] and for signaling [30].

Crude extracts from different plant species can hence contain a rich cocktail of bioactive compounds like alkaloids, essential oils, terpenes, sterols, and polyphenols [31] acting as elicitors, able to activate defense mechanisms in host plants against pathogens [9].

In order to consider the recent suggestion of the European Union in encouraging more sustainable waste management in agricultural fields (EN 491/2009) [31,32,33,46], four different agricultural/plant waste (grapevine stalk and marc, as well as *Ailanthus* and olive leaves) extracts were investigated. The experimental plan was designed to characterize their bioactive molecule content (Table 1) and their potential modulation of plant- and pathogen-related enzymes. Even though the heterogeneous complexity of secondary metabolites present in plant extracts makes it really difficult to identify the component responsible for biological activity, we focused our attention on polyphenols, a group of secondary metabolites widely recognized as modulators of plant biotic and abiotic stress response [9,20,34].

Our study demonstrated that the PEs of the considered agricultural/plant waste clearly showed a distinguished qualitative and quantitative polyphenol composition. In particular, we found a strong relationship between the PEs of quercetin-3-glycosides and *Ailanthus* and olive leaves, while flavonoids, such as procyanidin B1 and B3, taxifolin, catechin, epicatechin, and quercetin aglycone, were significantly associated with the PEs of grapevine stalks and marc.

In order to define the relationships between the PEs’ biological activity and their flavonoid composition, a first in vitro screening on their modulatory effects on purified enzymes involved in plant/pathogen interaction was performed (Figure 1 and Figure 2). As expected, PEs modulated differently the enzymatic activities involved in plant/pathogen interactions, as already stated by the study of Straney et al. [34], where host plant metabolites were used to trigger genes or developmental pathways needed for counteracting pathogenesis. In addition, Connolly and coworkers demonstrated the role of isoflavone exudates from soybean in triggering calcium influx into zoospores of *Phytophthora sojae*, affecting early steps of infection [35]. Moreover, Filippi and collaborators [13] recently demonstrated the role of quercetin and catechin as modulators of plant PR enzymes (chitinase, in particular), involved in biotic stress response.

In agreement with the recent evidence about the direct interaction between flavonoids and some plant PR proteins [36], in the present paper, we showed that the grape marc, as well as the PEs of *Ailanthus* leaves, induced the stimulation of both chitinase and peroxidase activity, respectively. Conversely, the pathogen enzymes commonly involved in plant biotic stress, such as cellulase, pectinase, amylase, and laccase, were mostly inhibited by grapevine marc, while no other PEs exhibited a large broad spectrum of inhibition.

The TOPSIS approach allowed identifying which is the best PE concentration able to enhance peroxidase and chitinase activity and concurrently to inhibit cellulase, laccase, pectinase, and amylase. Grape marc resulted the most effective PE, showing higher values at higher concentrations, suggesting that it could sustain/curb enzyme activity mostly because it contains high amounts of benzoates, cinnamates, and flavonols.

Among flavonols, syringetin derivatives, quercetin, and catechin have been already demonstrated to exert modulation of hydrolytic enzymes, such as alkaline phosphatase in human differentiating osteoblasts [37] and plant chitinase [13]. Vanillic and some *p*-hydroxybenzoic acids, given also to their strong antioxidant properties, have been demonstrated to stimulate the activity of antioxidant enzymes like catalase, peroxidase, superoxide dismutase, and polyphenol oxidase when applied to the germinating tomato plants [38] or to induce phytoalexin accumulation and enhance drought tolerance in rice [39]. In summary, we could only speculate that the above-mentioned compounds mostly represented in grape marc’s PE could cooperatively act as positive modulators of some defense-related enzymes likewise foliar chitinase, thus enhancing the response of plant during pathogen attack. These results could highlight the role of some phenolic compounds, largely accumulated in plant organs during biotic stress, not only as antioxidants per se but also as active molecules affecting the extent or efficacy of the plant immune responses against the pathogen.

From a future perspective, it would be essential to verify at the whole plant level if the modulation we measured on PR proteins might depend not only on the bioagent concentration but also on the exposure time and biodegradation processes of the extracts. These features would allow a long-lasting effect and, hopefully, prolonged absorption of the bioagents. For this purpose, it would be necessary to provide technical details and protocols that improve the persistence of the PEs, enabling adequate adhesion to the leaf surface and low leaching.

Actually, the data here presented opened the possibility that rough extracts obtained in agreement with the circular economy principle and characterized by several different chemical components might act, thanks to a synergic effect. Furthermore, the variegated composition of the plant extracts might allow to both stimulate the plant response liable for the defense against diseases (preventive effect) and possibly obtain at the same time an antibiotic action against pathogens (curative effect).

In addition, TOPSIS analysis returned 4 g L^−1^ as the most efficient PE dose able to modulate enzymes involved in host-pathogen interaction. This result is appreciable because it opens the possibility for the in vivo application of PEs, treating the whole plant even in the vineyard. In such a case, high PE concentrations could be required, considering their physicochemical degradation and loss during distribution. Consequently, the efficacy of PEs as an alternative strategy for pest management in the open field could be achieved as long as increasing the concentrations of PEs or the treatment frequency to counterbalance losses.

At a more complex level and considering the results of TOPSIS analysis, experiments on grapevine cell cultures were used as a suitable tool for studying the ex vivo effect of PEs on the activation of chitinases. In the cultured cells, a basal chitinase activity was detectable, which further increased after incubation with some PEs. In fact, several studies reported that cultured grapevine treated with elicitors of different origins, such as chitosan [40], methyl jasmonate [41], β-cyclodextrins [42], and laminarin [21], could produce typical defense responses, such as PR chitinases. Furthermore, a recent study on grapevine cells treated with a PE has elucidated the mechanism underlying plant defense elicitation [9].

Similarly, we found that grape marc extract at 4 g L^−1^ was effective in activating chitinases of grapevine cell culture (Figure 3A), similar to results obtained with purified chitinase from *Streptomyces* (Figure 1A). In both experiments, the peak of chitinase stimulation was reached at 4 g L^−1^. In a comparable way, a previous study found a similar correspondence in chitinase modulation induced by flavonoids [13]. The latter belongs to plant glycosidase family 19, exhibiting both endo- and exochitinase activity and higher expression in grape ripening berries [24].

According to these findings, we suggested that the efficacy of grape marc in stimulating cell-cultured chitinolytic activity could be mainly ascribed to the most quantitatively represented classes of flavonoids (i.e., the monomer catechin, dimeric procyanidins B2 + B4, and the flavonols quercetin and syringetin- glucoside and galactoside). Recently, 4-hydroxycoumarin derivatives have been selected for their predicted highly binding affinity for fungal chitinase and considered as new potential antifungal inhibitors [25]. It is conceivable that some, if not all, of these phenolic molecules, could directly exert their stimulation on the chitinase activity measured in grapevine cells, depending on their concentration and affinity to the catalytic site. Nonetheless, we could not exclude *a priori* that other chitinases than type IV could be present and active in the studied grapevine cell cultures, although previous studies reported this class as the major represented in grape [24]. Like other PR plant proteins, chitinases are generally synthesized in response to biotic or abiotic stresses [43], and indeed some of them, including type IV, cause effective fungal growth inhibition [24]. Moreover, class IV chitinases could also potentially promote programmed cell death, as some authors reported that the *EP3* class IV chitinase genes in *Arabidopsis* are similarly involved in development-related apoptotic processes [44]. Furthermore, several recent studies have described that pepper class IV chitinases could regulate basal resistance and hypersensitive-like programmed cell death during infection with avirulent bacterial pathogens, through interaction with a receptor-like cytoplasmic protein kinase CaPIK1 [45] or with the fungal pathogen *Phytophthora capsici* [47]. Interestingly, in our model with grapevine cell cultures, we found that stimulation of chitinase activity by grape marc’s PE was also associated with a parallel increase in cell death, although without statistical significance (Figure 3B).

A further confirmation that grape marc’s PE modulated chitinase activity in grapevine plants was obtained by in vivo experiments, where foliar chitinase was greatly stimulated by the latter in a concentration and time-dependent manner (Figure 4). Our results might indirectly confirm what has been reported in several genomic analyses, where overexpression of chitinase genes was induced by different elicitors [9,22]. In our study, the grapevine leaves, harvested from the treated plants, were thoroughly washed before enzymatic extraction and the chitinase was activated in a time-dependent manner, so we could arise two considerations: i) some specific phenolic compounds of grape marc’s PE possess a potential action in eliciting overexpression of defense proteins likewise chitinases; ii) otherwise, as alternative, these compounds could exert their direct action by interacting with apoplastic chitinases, thus modulating their enzymatic activities. The second hypothesis, however, could not fully explain the different extent of chitinase activation observed at 8, 24, or 48 h after treatment, since a direct modulation of chitinase predicts that its action could be dependent mainly on concentration and not on time of exposure.

Further investigation is necessary to understand whether the most active polyphenols, such as those mainly enriched in the grape marc extract, could exert their modulation action on chitinase and other PR proteins, synergistically with a direct biocide effect against fungal pathogens. This approach would need validation through the in vitro fungal growth/germination inhibition test, which was, however, beyond the scope of the present research. These analyses and the obtained results here presented would open future perspectives regarding the evaluation of antimicrobial activity exerted in the vineyard by PEs. In particular, it would be necessary to measure in the open field the efficacy of the most promising PEs, likely grape marc, against the most widespread grapevine pathogens, in order to understand whether their application could concur to decrease the need for chemical pesticide treatments.

## 4. Materials and Methods

### 4.1. Bioactive Compound Extraction

Four different hydro-alcoholic PEs were obtained from different plant materials—olive leaves (*Olea europea* L.); the tree of heaven leaves (*Ailanthus altissima* (Mill.) Swingle); grape marc (*V. vinifera* L., cv. Refosco dal peduncolo rosso); grape stalks (*V. vinifera* L., cv. Refosco dal peduncolo rosso)—harvested at local places and stored at −20 °C until extraction. For all materials, 12 g (FW) of frozen material was ground into powder under liquid nitrogen in a frozen kitchen mixer and resuspended into 30 mL (i.e., 400 g L^−1^) of hydro-alcoholic solution (50%, v/v). Incubation was undertaken at 4 °C for 30 min under stirring. After centrifugation at 28,000 × *g* for 10 min, the supernatant of each PE was collected and stored at −20 °C.

### 4.2. Analysis of Polyphenols by Means of UHPLC-MS

Phenolic compounds were determined with a method adapted from Vrhovsek and coworkers [48], where also limit of quantification (LOQ) values for every single compound were reported. Limit of detection (LOD) value was calculated as 1/3 of the LOQ value. The extract was filtered through a 0.2 μm polytetrafluoroethylene (PTFE) filter prior to liquid chromatography. Chromatographic analysis was performed using a Waters Acquity UPLC system (Milford, CT, USA) with a Waters Acquity HSS T3 column (100 mm × 2.1 mm; 1.8 µm). Mass spectrometry detection was performed on a Waters Xevo triple-quadrupole mass spectrometer detector (Milford, CT, USA) with an electrospray ionization (ESI) source [48]. Compounds were identified based on their reference standard, retention time, and qualifier and quantifier ion and were quantified using their calibration curves and expressed as mg/kg of fresh leaves. Data processing was performed using Waters MassLynx V4.1 software (Milford, CT, USA) [48].

### 4.3. Cell Culture Suspension and Plant Material: Growth and Maintenance

Grapevine cell culture (*V. vinifera* L., cv. Cabernet Sauvignon) was a generous gift of Prof. Stéphanie Cluzet (ISVV-UR Enology, University of Bordeaux, France). Cell suspension culture was maintained in B5 medium supplemented with 20 g L^−1^ sucrose, 250 mg L^−1^ casein hydrolysate, 0.5 mg L^−1^ naphthaleneacetic acid, and 0.12 mg L^−1^ benzylaminopurine [49]. Suspension culture was weekly sub-cultured in 250 mL Erlenmeyer flasks containing 100 mL of cell suspension, by inoculating the cells at a 1/5 (v/v) ratio into fresh medium.

One-year-old rooted cuttings of *V. vinifera* L. cv. Verduzzo Friulano were grown in 1 L plastic pots containing perlite and topsoil at standard conditions (25 °C, 60% humidity, light/dark cycles of 14/10 h, the light intensity of 4000 lux).

### 4.4. Cell Suspension Cultures (Ex Vivo) Treatment by Bioactive Compounds

Three-day-old cell suspension cultures were exposed to each PE (4 g L^−1^ as final concentration) or an equivalent volume of ethanol/water 50% v/v in control for 48 h.

### 4.5. Plant (In Vivo) Treatment by Bioactive Compounds

At the stage of the 7th completely expanded leaf, adaxial and abaxial leaf surfaces of three plants per each treatment were sprayed with the grape marc’s PE, at two different concentrations (4 and 800 g L^−1^) dissolved in 5 mL of 0.1% coadjuvant (Etravon, Syngenta, Italy). An equal volume of a water/alcohol solution with coadjuvant was also used as control. The concentration of 800 g L^−1^ was obtained by lyophilizing an aliquot of 400 g L^−1^ solution and resuspending it in a half volume.

The exposure time with the PE was scheduled for 8, 24, and 48 h.

### 4.6. Cell Death Determination in Cell Suspension Cultures

To determine the extent of PE-induced cell death, the aliquot of suspension culture was incubated with fluorescein diacetate for 5 min in the dark. The cell death ratio was calculated as the number of dead cells divided into all the cells (living cells recognizable due to green fluorescence emission), monitored under the fluorescent microscope (LEICA Fluovert, Wetzlar Germany).

### 4.7. Modulation of In Vitro Pathogen-Related and Plant Defense-Related Enzyme Activities

#### 4.7.1. Purified Enzymes; In Vitro Modulation

Four PEs obtained from different plant species and tissues (*Ailanthus* leaves, grape marc, grape stalks, and olive leaves), known to be enriched in bioactive compounds (particularly polyphenols), were evaluated for in vitro modulatory effect of plant defense-related enzymes, including cellulase, pectinase, amylase, laccase, chitinase, and peroxidase. The analyses were carried out using the different purified commercial enzyme solution and incubated with different concentrations of the aforementioned PEs (0.4, 4, and 40 g L^−1^) for 15 min at 25 °C, before starting the reaction in the presence of the related substrate of each enzyme. Incubation mixture used for the specific enzyme assay was as follows:

i) The 3 U mL^−1^ of purified cellulase enzyme (EC 3.2.1.4) obtained from *Trichoderma longibrachiatum* was pre-incubated with different PE concentrations in 50 mM sodium acetate buffer at pH 4.8 for 15 min. Then, 450 µg of cellulose as the substrate was added to the solution, and the mix solution was incubated for another 2 h at 50 °C. Subsequently, 150 µL of 5% of 3,5-dinitrosalicylic acid (DNSA) solution was added to 100 µL of the mix solution and incubated at 90 °C per 10 min. After cooling, the absorbance of the samples was read at 570 nm through Multilabel Counter (WALLAC, model 1420, Perkin-Elmer, Waltham, MA, USA) [50];

ii) The 0.24 U mL^−1^ of purified pectinase enzyme (EC 3.2.1.15) with the origin of *Aspergillus niger* was pre-incubated with different PE concentrations in 50 mM sodium acetate buffer at pH 4.8 for 15 min. Then, 7.5 µg of pectin was added to the solution and incubated for 1 h at 37 °C. Subsequently, 100 µL of 5% of the DNSA solution was added to the mix solution and incubated at 90 °C per 10 min. After cooling, the absorbance of the samples was read at 570 nm by Multilabel Counter (Perkin-Elmer, Waltham, MA, USA) [51];

iii) The 750 U mL^−1^ of purified amylase enzyme (EC 3.2.1.1) isolated from *Bacillus* sp. was pre-incubated with different PE concentrations in 20 mM sodium phosphate buffer at pH 6.9 for 15 min. Then, 250 µg of starch was added to the solution and incubated for 2 h at 25 °C. Subsequently, 100 µL of 5% of the DNSA solution was added to the mix solution and incubated at 90 °C per 10 min. After cooling, the absorbance of the samples was read by a Multilabel Counter (Perkin-Elmer, Waltham, MA, USA) set at 570 nm and 465 nm (20 nm excitation/emission filter bandwidth) [52];

iv) The 125 U mL^−1^ of purified laccase enzyme (EC 1.10.3.2) obtained from *Aspergillus* sp. was pre-incubated with different PE concentrations in 100 mM sodium phosphate buffer at pH 6.5 for 15 min. Then, 0.1% of guaiacol was added to the solution and incubated for an hour. The absorbance was read by a Multilabel Counter (Perkin-Elmer, Waltham, MA, USA) set at 490 nm and 465 nm (20 nm excitation/emission filter bandwidth) [53];

v) The 0.85 mU mL^−1^ of purified chitinase enzymatic activity (EC 3.2.1.14) obtained from *Streptomyces griseus* was pre-incubated with different PE concentrations in McIlvain’s buffer (43 mM citric acid and 114 mM sodium phosphate dibasic pH 5.5), measured as described by Filippi et al. [13]. The reaction started in the presence of the substrate 4-methylumbelliferyl β-d-N,N′,N′-tri-acetyl-chitotrioside (Sigma Aldrich, Milan, Italy), at a final concentration of 10 μM with McIlvain’s buffer, pH 5.5. The reaction required a minimum of 2 h of incubation at 37 °C to be completed. The fluorescence of the product (4-methylumbelliferone) released by enzymatic hydrolysis was measured utilizing a Multilabel Counter (WALLAC, model 1420, Perkin-Elmer, Waltham, MA, USA) set at 340 nm and 465 nm (20 nm excitation/emission filter bandwidth). Chitinase activity was calculated as RFU µg^−1^ prot. min^−1^ or as a percentage of each control, respectively.

vi) The 80 mU mL^−1^ of purified peroxidase enzymatic activity (1.11.1.7) from horseradish type I was pre-incubated with different PE concentrations in McIlvain buffer for 15 min. Then, 2.2 mM of hydrogen peroxide was added, and the absorbance was read after 30 min of incubation by Multilabel Counter (WALLAC, model 1420, Perkin-Elmer, Waltham, MA, USA) set at 450 nm and 465 nm (20 nm excitation/emission filter bandwidth) [54].

#### 4.7.2. Chitinase Activity; Ex Vivo Modulation in *V. vinifera* cv. Cabernet Sauvignon Suspension Cultures

Forty-eight hours after incubation, aliquots of suspension cell cultures were filtered through a 50 µm nylon gauze, briefly washed with the same medium and immediately ground into frozen powder under liquid nitrogen. Aliquots (250 mg, FW) of powder were homogenized in extraction buffer (0.2 M sodium acetate buffer, pH 5.5, with 5 mM dithioerythritol) in a 1:2 ratio. After centrifugation at 14,000× *g* for 10 min at 4 °C, the supernatants were used for both protein assay (Bradford Reagent, Sigma-Aldrich, Milan, Italy) and chitinase activity determination, as described above. Chitinase activity was calculated as RFU (h x µg protein)^−1^, and then the value for each treatment was compared to their related control.

#### 4.7.3. Chitinase Activity: In Vivo Modulation in *V. vinifera* cv. Verduzzo Friulano Plants

The third and fourth leaves from the shoot tip were collected at the course time of 8, 24, and 48 h after treatment application. Leaves were washed up thoroughly with distilled water to remove PE residue left over the cuticle. Then, they were ground into a fine powder under liquid nitrogen and stored at −80 °C. The powder was re-suspended in the extraction buffer (200 mM sodium acetate pH 5.5, 5 mM DTE and 0.1% CHAPS) in a 1:2 ratio, homogenized with potter and pestle, and centrifuged at 14,000× *g* for 10 min. The supernatant was collected and used for both protein assay and determination of the enzymatic activity, as described previously.

### 4.8. Statistical Analysis

The experiments were conducted with at least three replicates in all cases, except for the cell culture assays, where seven replicates were carried out. All data are presented as mean ± standard deviation (SD). The effects of the different PEs on purified enzymes and suspension cell cultures were statistically analyzed using a one-way ANOVA test. Data obtained from cutting treatments were instead analyzed by a two-way ANOVA test. After ANOVA, means were compared by the post hoc test of least significant difference (LSD) analysis, according to Fisher’s statistical test, when the normal data distribution was verified. A non-parametric Kruskal–Wallis test was applied in case of ANOVA model assumption violation. The difference between the means was considered significant when the probability coefficient was at *p* ≤ 0.05.

In order to assess the putative best bioactive compounds, in relation to its concentration, we applied multi-criteria decision analysis. We used the TOPSIS method (a technique for order preference by similarity to an ideal solution) [55,56], which uses the basic principle that the best solution should have the shortest distance from the ideal solution and the farthest distance from the negative ideal solution. In this light, we used the increase (max values) of peroxidase and chitinase enzyme activity and the decrease of cellulase, laccase, pectinase, and amylase (min values) as the ideal solution and their lowest and highest values, respectively, as the worst solution. The indices provide a value ranging between 1, if and only if the alternative solution has the best condition, and 0, if and only if the alternative solution has the worst condition.

The composition of the bioactive polyphenolic profile performed by HPLC-MS was analyzed by applying a principal component analysis (PCA). This analysis was performed in R statistical software [57].

## Figures and Tables

**Figure 1 ijms-20-06357-f001:**
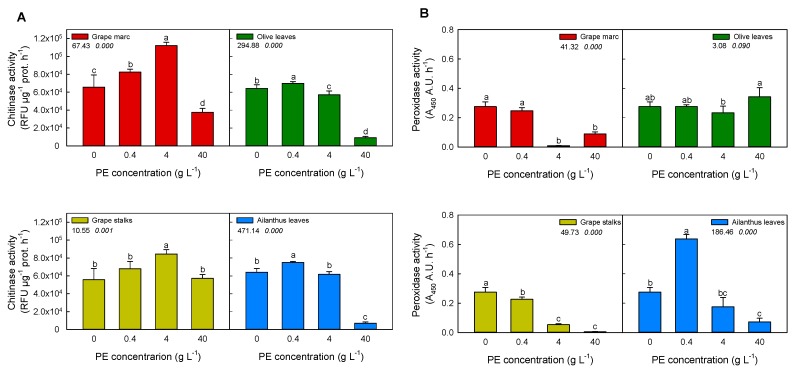
Effect of different concentrations of plant extracts (PEs) on plant defense-related enzyme activities. Bacterial chitinase (**A**) and horseradish peroxidase (**B**) activities were determined after pre-incubation with 0, 0.4, 4, and 40 g L^−1^ of PE, respectively. PEs were pre-incubated with purified enzymes in each assay buffer at different concentrations for 15 min before starting the reaction, as described in Materials and Methods. Different superscript letters indicate significant difference (*p* ≤ 0.05) as assessed by one-way ANOVA or non-parametric Kruskal–Wallis test, followed by the relative post-hoc test. Values under the treatment legend represent F and *p* values, respectively, obtained by ANOVA analysis.

**Figure 2 ijms-20-06357-f002:**
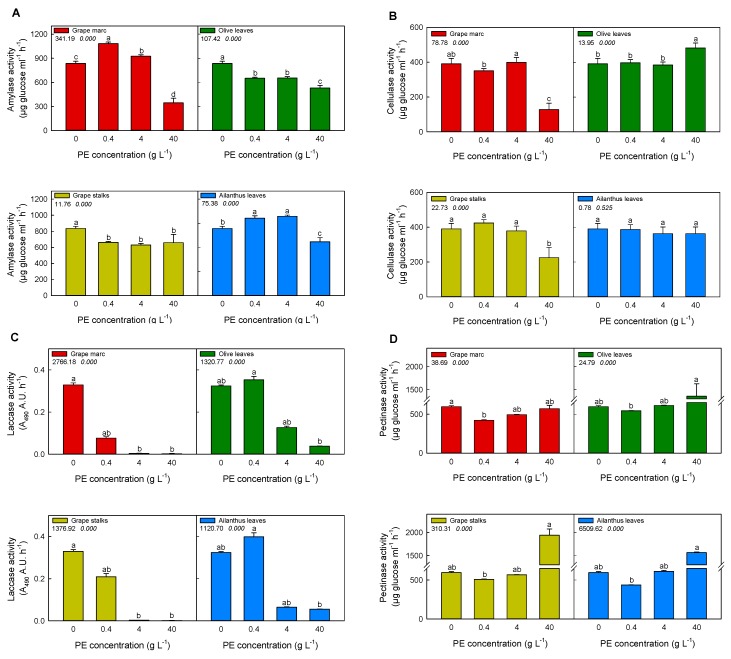
Effect of different concentrations of PEs on pathogen-related enzyme activities. Bacterial amylase (**A**), fungal cellulase (**B**), fungal laccase (**C**), and fungal pectinase (**D**) activities were determined after pre-incubation with 0, 0.4, 4, and 40 g L^−1^ of PE, respectively. PEs were pre-incubated with purified enzymes in each assay buffer at different concentrations for 15 min before starting the reaction, as described in Materials and Methods. Different superscript letters indicate significant difference (*p* ≤ 0.05) as assessed by one-way ANOVA or non-parametric Kruskal–Wallis test, followed by the relative post-hoc test. Since the normal distribution of data was not verified, in the statistical analysis of pectinase and laccase activities, a non-parametric Kruskal–Wallis test was applied. Values under the treatment legend represent F and *p* values, respectively, obtained by ANOVA analysis.

**Figure 3 ijms-20-06357-f003:**
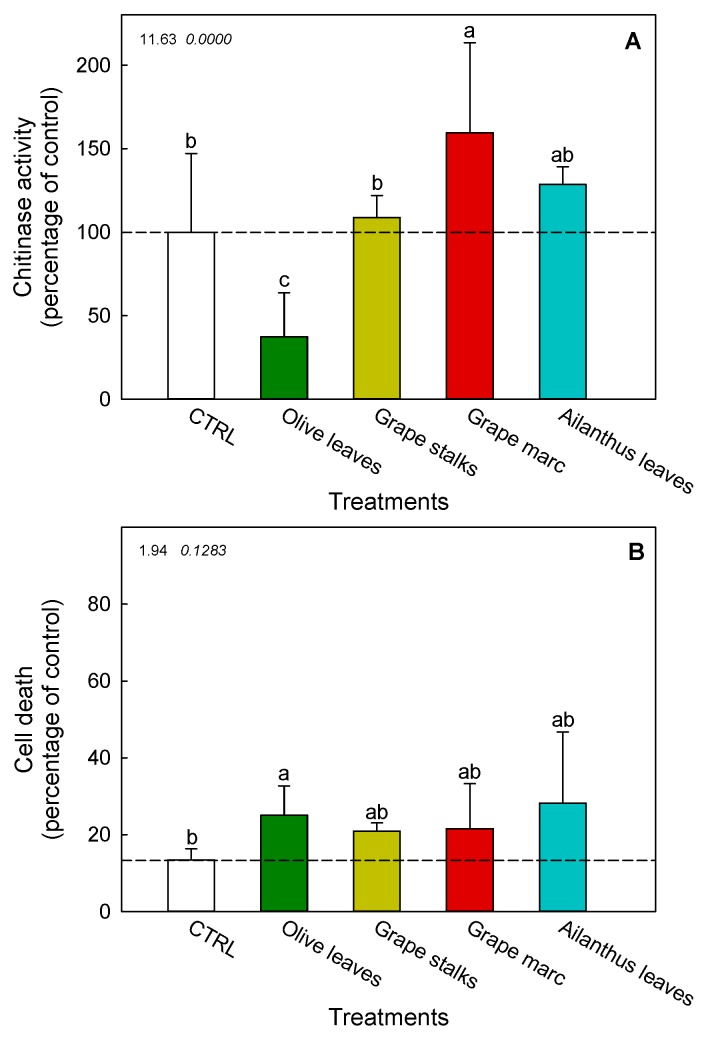
Effect of different PEs on the modulation of chitinase activity in grapevine cv. Cabernet Sauvignon cell suspension cultures. Chitinase activity (**A**) and cell death (**B**) were determined after pre-incubation of three-days-old grapevine suspension cultures for 48 h with PEs at the concentration of 4 g L^−1^. Different superscript letters indicate significant difference (*p* ≤ 0.05) as assessed by one-way ANOVA or non-parametric Kruskal–Wallis test, followed by the relative post-hoc test. Since the normal distribution of data was not verified, in the statistical analysis of cell death, a non-parametric Kruskal–Wallis test was applied. Values inside the plot represent F and *p* values, respectively, obtained by ANOVA analysis.

**Figure 4 ijms-20-06357-f004:**
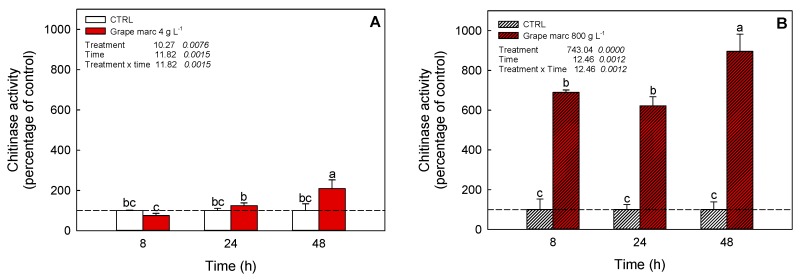
Effect of different concentrations of grape marc’s PE on the modulation of chitinase activity in grapevine cv. Verduzzo Friulano leaf supernatants. Grape marc’s PE was sprayed on leaves of one-year-old grapevine plants at the concentration of 4 (**A**) and 800 g L^−1^ (**B**), and leaf cellular chitinase activity was measured after 8, 24, and 48 h after treatment, as described in Materials and Methods. Different superscript letters indicate significant difference (*p* ≤ 0.05) as assessed by one-way ANOVA or non-parametric Kruskal–Wallis test, followed by the relative post-hoc test. Values under the treatment legend represent F and *p* values, respectively, obtained by ANOVA analysis.

**Figure 5 ijms-20-06357-f005:**
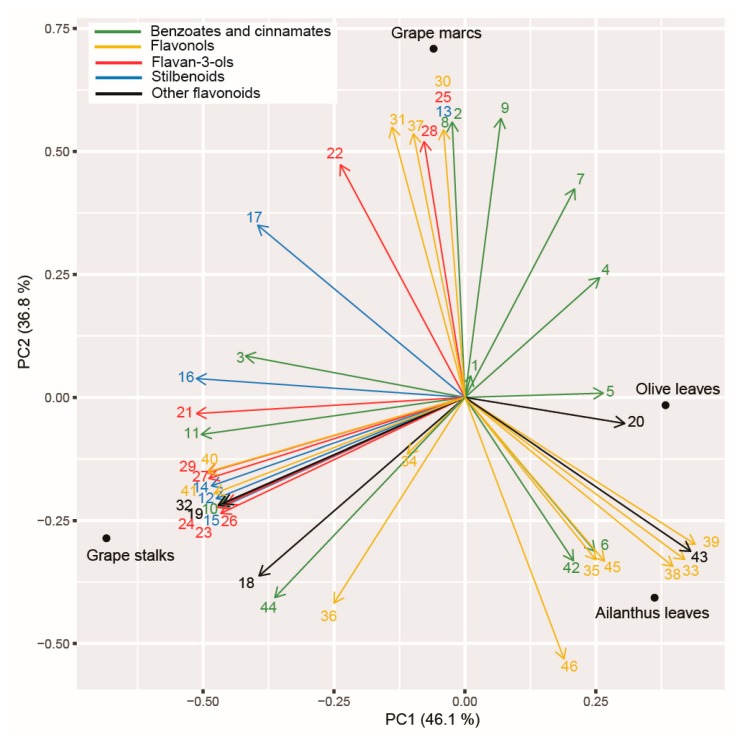
Biplot of the principal component analysis (PCA) of the polyphenolic profile of the four assayed PEs. The first two components are shown. Polyphenolic compounds’ codes are consistent with those shown in Table 1.

**Table 1 ijms-20-06357-t001:** Polyphenolic profile of the plant extracts (PEs) obtained by HPLC-MS analysis. All PEs were obtained by hydro-alcoholic extraction starting from 400 g L^−1^ fresh weight (FW) of plant material. Limit of quantification (LOQ) is also indicated.

Polyphenols	Grape Marc (mg L^−1^)	Grape Stalks (mg L^−1^)	Olive Leaves (mg L^−1^)	*Ailanthus* Leaves (mg L^−1^)	LOQ (pg)	Reference Number
***Benzoates and cinnamates***						
4-aminobenzoic acid	<L.O.Q.	<L.O.Q.	-	-	n.a.	-
*p*-hydroxybenzoic acid	0.11	0.19	0.37	-	40	[1]
vanillic acid	0.81	0.02	0.09	-	20	[2]
gallic acid	4.07	4.79	-	2.58	160	[3]
3,5-dihydroxy-benzoic acid	0.17	-	0.38	-	16	[4]
protocatechuic acid	0.11	-	2.89	-	80	[5]
methyl gallate	-	<L.O.Q.	-	1.21	8	[6]
*p*-coumaric acid	0.13	-	0.15	-	12	[7]
caffeic acid	0.16	-	-	-	12	[8]
ferulic acid	0.23	0.01	0.11	-	2	[9]
caftaric acid	0.36	33.41	0.80	0.63	120	[10]
fertaric acid	0.24	0.78	0.10	0.03	40	[11]
ellagic acid	1.65	5.51	-	53.79	800	[24]
***Stilbenoids***						
*t*-resveratrol	0.07	2.36	-	-	224	[12]
*cis*-resveratrol	0.02	-	-	-	8	[13]
piceatannol	0.07	0.86	-	-	12	[14]
*t*-piceid	0.05	0.79	-	0.07	15	[15]
*cis*-piceid	0.16	0.34	-	<L.O.Q.	8	[16]
isorhapontin	0.05	0.04	-	-	8	[17]
***Chalcones***						
phlorizin	0.03	0.36	0.03	0.16	4	[18]
***Flavones***						
sinensetin	-	0.01	-	-	160	[19]
luteolin	0.02	-	10.94	1.40	200	[20]
luteolin-7-O-Glc	0.19	0.24	23.86	31.86	12	[25]
*Flavan-3-ols*						
catechin	16.30	46.97	<L.O.Q.	0.04	20	[21]
epicatechin	13.35	5.50	-	0.91	400	[22]
epigallocatechin	1.05	3.44	-	1.40	16	[23]
gallocatechin	0.82	36.70	-	0.94	20	[26]
catechin gallate	0.07	-	-	-	20	[27]
epicatechin gallate	0.14	8.05	-	0.50	2	[28]
procyanidin B1	7.71	68.93	-	0.10	120	[29]
procyanidin B2 + B4	358.93	44.43	-	38.34	80	[30]
procyanidin B3 (as B1)	4.19	30.77	-	0.14	20	[31]
***Flavonols***						
kaempferol	0.27	-	-	-	8	[32]
quercetin	3.46	1.36	1.46	-	8	[33]
quercetin-3-Rha	0.01	0.24	5.49	7.87	8	[34]
myricitrin	0.02	0.02	-	0.03	20	[35]
quercetin-3-Glc + quercetin-3-Gal (as que-3-Glc)	0.32	5.59	7.03	142.94	12	[36]
isorhamnetin-3-Glc	0.14	0.43	0.10	0.39	8	[37]
syringetin-3-Glc + syringetin-3-Gal (as syr-3-Glc)	2.68	0.31	-	-	8	[38]
rutin	0.08	2.41	39.42	71.42	16	[39]
quercetin-3,4-diglucoside	0.06	0.05	0.80	0.95	20	[40]
quercetin-3-glucuronide	2.27	16.11	0.02	0.12	40	[41]
kaempferol-3-glucuronide	0.03	0.59	-	-	8	[42]
arbutin	-	0.34	0.07	0.14	200	[43]
kaempferol-3-rutinoside	-	0.31	0.87	9.89	4	[44]
isorhamnetin-3-rutinoside	-	0.31	0.35	0.58	8	[45]
***Flavanonols***						
taxifolin	0.30	6.19	0.71	0.16	12	[46]

**Table 2 ijms-20-06357-t002:** Calculated TOPSIS (technique for order preference by similarity to an ideal solution) values for the PEs obtained by in vitro plant defense- and pathogen-related enzyme activities.

Agent	0.4 (g L^−1^ FW)	4 (g L^−1^ FW)	40 (g L^−1^ FW)	Mean
*Ailanthus* leaves	0.51	0.50	0.39	0.46
Grape marc	0.53	0.54	0.64	0.57
Olive leaves	0.49	0.54	0.45	0.49
Grape stalks	0.51	0.57	0.48	0.52
Mean	0.51	0.54	0.49	0.51

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
