# Peer review of "Bioactive Polyphenols Modulate Enzymes Involved in Grapevine Pathogenesis and Chitinase Activity at Increasing Complexity Levels"

_ijms, 2019, doi:10.3390/ijms20246357_

Round 1

Reviewer 1 Report

Plant cultivation is associated with the risk of pathogens and pests. Synthetic substances, although effective, unfortunately proved to be harmful to humans and animals.
Therefore, for over two decades, global trends have focused on the search for harmless and effective natural remedies for the use of synthetic substances. It has long been known that polyphenols present in plants have antibacterial and antiviral activity. These properties are successfully used to reduce plant pathogens. The manuscript presented for review is in line with these trends, by studying the effect of hydroalcoholic plant extracts from pomace Vitis vinifera L. and leaves of Olea europaea L. and Ailanthus altissima (Mill.) Swingle) on chitinase activity and plant-pathogen interactions. The authors presented the analysis of raw materials in terms of chemical composition and in vitro studies confirmed in cultivation conditions regarding chitinase activity, which is modulated by polyphenols. However, it is difficult to clearly determine which substances had a greater impact. Or maybe the obtained effect in relation to the vine extract is due to the synergism of phenolic acids, flavonoids and stilibenes. Potential activity in overexpressing protein defense may result from the concentration of the substance but also from the time of exposure.

The research methodology is appropriate and thoroughly described. The authors discuss the results obtained in detail and put forward theses that can explain their results.

On the other hand, the list of references is too extensive, which could be limited to the latest reports.

Author Response

We are grateful to the reviewer for his meaningful observations. In agreement with his suggestions, we add to the discussion a paragraph explaining, at least at theoretical level, which polyphenol could be considered as the most effective compound exerting stimulation on the plant defence enzymes. Indeed this effect could be due to a synergic action resulting from the heterogeneous composition of the extracts, as well as from specific active molecules. Such molecules may function per se modulating enzymatic activities, thanks to their chemical structure or as elicitors of de novo synthesis of the PR enzymes, and finally they even exhibit an antimicrobial activity. In agreement with these premises, the proposed use of bioagents in open field treatments would necessarily consider also an adequate number of applications to improve their persistence on plant surface, in term of long lasting adhesion and low degradation. 

Accordingly to reviewer observations, we have also reduced the list of references, by deleting the oldest citations.

Reviewer 2 Report

Comments:

Scientific names should be revised to be Italic

References and reference list also to be revised in line with the author guidelines.

English language to be revised by proofreader or by native English speaker.

The manuscript to be revised for the typographic errors and spaces.

Other comments:

I was wondering about the optimization method for the different phenolic compounds and the parameters used in the quantitation process. In order to be acceptable, all parameters used should be included into the text together with the LOQ and LOD values.

Author Response

We thank the reviewer for his observations and accordingly we have improved the manuscript. In particular, we revised scientific names by formatting in italic style and we check the reference list according to the author guidelines. We amended typographic errors and the English language was revised by a proofreader.

We insert a column in Table 1, reporting LOQ values for the polyphenol profile. We add also a phrase with reference to a previous paper, describing with more detail how these parameters were obtained.

Round 2

Reviewer 2 Report

Now, the Ms has been improved and can be accepted for publication in the IJMS